# Stress and Coping Behavior Exhibited by Family Members Toward Long-Term Care Facility Residents While Hospitalized

**DOI:** 10.3390/healthcare12202022

**Published:** 2024-10-11

**Authors:** Han-Lin Kuo, Yi-Wen Chiu

**Affiliations:** 1Department of Nursing, Show Chwan Memorial Hospital, Changhua 50008, Taiwan; piano1506652003@yahoo.com.tw; 2Department of Nursing, Chung Shan Medical University, Chung Shan Medical University Hospital, Taichung 40201, Taiwan

**Keywords:** long-term care facility, family members, stress and coping behavior

## Abstract

Background: With the increase in the elderly population, institution-based care has become another option for elderly people. In Changhua, Taiwan, the number of long-term institutions has doubled in the past decade, and more families are choosing to send their elders to institutions for care. However, there is stress induced by having to care for these elders when they come back to their family members when hospitalized. Therefore, this study aimed to understand the stress and coping behaviors of family members in regard to hospitalized long-term care facility residents and identify relevant factors that affect and predict the stress and coping behaviors exhibited by these family members. Method: In this study, a quantitative and cross-sectional survey was conducted using the convenience-sampling method; family members of long-term care facility residents hospitalized in a regional hospital in central Taiwan were selected as the research participants and a total of 162 family members were admitted. The data were collected in the form of questionnaires including basic information and data on the stress and coping behaviors of the family members. The data were collected and coded by using SPSS 22.0 to perform descriptive and inferential statistical analysis. Results: The standard average score of total stress for family members was 57.03 points, which corresponds to a moderate level. The four perceptions of stress by family members were, in order, physiological, life, psychological, and economic. Furthermore, family income, work status, and the relationship between residents of the family members and physiological, psychological, and economic factors had predictive power for their problem-oriented coping behaviors, with an explanatory power of 59.6%. Life aspects, gender, marital status, and the number of hospitalizations in half a year had significant predictive power for the family members’ emotion-oriented coping behaviors, with an explanatory power of 19.0%. Conclusions: The family members had high levels of stress, especially physical stress, and the total scores of stress perception were higher for those who were younger than 39 years old and had no rotating family members. Additionally, the coping behavior of the main caregiver was mainly problem-oriented. The results of this study may serve as a reference that can help nursing staff in clinical or long-term care facilities to provide or develop effective and individualized services for family members of facility residents.

## 1. Introduction

With the advancement of medical technology, the average life expectancy has increased. In Taiwan, in 1993, there were 1.49 million people over 65, representing a ratio of 7% of the total population, marking the onset of an aging society. By 2018, the number of people over 65 had reached 3.32 million, accounting for 14.09% of the total population, signifying a transition to an aged society [1]. Compared with countries around the world, which typically take at least 72 to 125 years to transition from an aging society to an aged society, Taiwan accomplished this in just 25 years, making it the country with the fastest aging population. As of 2021, Taiwan’s older people over 65 represented 16.85% of the total population, and it is estimated that, by 2026, Taiwan’s older population will exceed 20.7%, meaning one in five individuals will be older people, marking the onset of a super-aged society [1,2].

In 2050, Taiwan’s total population will experience negative growth, with a natural increase rate of −10.60‰. The proportion of people aged 65 and above in the total population will be 36.60%, with an old-age dependency ratio of 67.70% and an aging index as high as 393.50% [3]. In 1991, the average household size was 3.94 people, but by the end of 2022, it had decreased to 2.59 people. This reduction in average household size signifies a decline in the functionality of families [1]. In response to the challenges posed by an aging society, the government is vigorously promoting community- and home-based care to enable the elderly to receive care in their familiar surroundings. However, as times change, household sizes are gradually decreasing, and the functionality of families is diminishing, leading to practical considerations. In situations where the elderly experience physical decline and are unable to care for themselves and nuclear families struggle to balance work with the dual role of caring for older people, institutional care has emerged as a viable alternative. Families find themselves compelled to seek long-term care facilities to provide support for the elderly in their later years [4,5,6].

Statistics from the Ministry of Health and Welfare reveal that there were 328 long-term care institutions added in Taiwan from 2007 to 2017, marking a 25.5 percentage point increase compared to 2007 [7]. This growth underscores the increasing demand for institutional long-term care for the elderly, a reality that cannot be ignored [8,9]. Long-term care service institutions not only have professional care staff and volunteers but also provide 24 h technical, life, and accommodation services for older people living with disabilities or dementia. They offer continuous healthcare tailored to residents’ health conditions and self-care abilities, providing individualized daily care in regard to physical, psychological, social, and spiritual aspects. This aims to maintain residents’ physical functions and enhance their independent living capabilities, allowing them to live in a respectful and caring environment. It is evident that institutions not only offer professional care quality but also alleviate the burden and stress on adult children or family caregivers in the caregiving process [10,11,12,13].

However, most studies addressing caregiver stress focus on the stress experienced by caregivers within the family setting [14,15,16,17]. There is limited research on the stress experienced by the family members of residents in long-term care institutions. Does the placement of older individuals in long-term care institutions completely relieve their family members of caregiving stress? Are there other situations that may impose greater caregiving stress on them? These are key concerns for researchers.

Placing an elderly individual in an institution does not signify the end of caregiving responsibilities for family members. The emotions of family members may be complex as they actively engage with the institutional life of the elderly [18]. However, because family members often work in different locations, receiving notifications from long-term care institutions about residents’ hospitalization may lead to varying levels of stress, particularly when balancing caregiving responsibilities with conflicts in life and work. When the elderly are hospitalized, families not only have to cover the original expenses but also incur additional hospitalization costs. Although hospitals have “cooperation beds” arranged with long-term care institutions, meaning four patients share the cost of one caregiver, this feature is still an additional expense for families. Therefore, we were motivated to explore the stress and coping behaviors of family members taking over caregiving responsibilities when residents in long-term care institutions are hospitalized.

## 2. Materials and Methods

In line with the above research objectives, this study aims to address the following research questions:(1)What is the level of stress experienced by family members when their loved ones in long-term care facilities are hospitalized?(2)What coping behaviors do family members employ when their loved ones in long-term care facilities are hospitalized?(3)How do various stress factors influence the coping behaviors of family members when their loved ones in long-term care facilities are hospitalized?(4)What are the predictive factors associated with the stress levels and coping behaviors of family members when their loved ones in long-term care facilities are hospitalized?

To address the aforementioned research questions, this study adopted a quantitative, cross-sectional survey method, targeting family members of long-term care facility residents hospitalized in a regional hospital in central Taiwan Province. The participants were family caregivers of residents hospitalized in long-term care facilities from August 2021 to April 2022. The inclusion criteria for participants were family members aged 20 and above, possessing clear consciousness, and lacking language or text communication barriers. A total of 162 family members were included in the study. Data collection was conducted through face-to-face interviews by the same researcher. Considering the family members might be anxious about their loved ones’ illness and initial interactions with strangers might be challenging, efforts were made to establish a trusting and respectful rapport with the interviewees. Interviews were conducted within 48 h of the patients’ hospitalization. The researcher emphasized the confidentiality of personal information and assured participants that their rights and privacy would be protected. Additionally, interviews were conducted in a controlled environment without the presence of third parties to minimize the impact of socially desirable responses.

The research tools for this study include three parts: basic personal information, a stress scale, and a coping behavior scale. The stress scale was independently developed by the researcher based on references in the literature, focusing on four dimensions: “physiological” (e.g., insomnia, poor health), “psychological” (e.g., bad mood), “economic” (e.g., expenses for additional hospitalization or hiring of caregivers), and “life” (e.g., the pressure of taking time off and interference in life) with a total of 20 questions. A Likert scale was used, employing a five-point scoring method (total score: 20–100), where higher scores indicate greater stress experienced by family members. The coping behavior scale was developed by Lin in 1996, with authorization, and has good reliability and validity [19]. This scale consists of 25 questions, scored using a Likert scale, where higher scores indicate better coping ability of family members (total score range: 25–125).

After the completion of the scales, content validity was verified by experts. Five experts in the field of long-term care were invited to provide professional opinions on the appropriateness and clarity of the questionnaire content. Each question was rated on a scale of 1 to 5 to assess its validity, and the CVI (Content Validity Index) was calculated. The CVI for the stress scale in this study was 0.98, and for the coping behavior scale, it was 0.89. Regarding reliability, internal consistency testing was adopted, and the overall Cronbach’s alpha value for the scales was 0.724. The Cronbach’s alpha values for the different dimensions of the stress scale were as follows: physiology (0.734), psychology (0.705), economy (0.873), and life (0.490). The Cronbach’s alpha value for the coping behavior scale was 0.733, with emotional coping at 0.904 and problem coping at 0.594 (As proposed by Nunnally in 1967, Cronbach’s alpha values between 0.35 and 0.7 indicate that the scale has good reliability and validity [20]).

This study was conducted following the review and approval of the Research Ethics Review Committee of the Show Chwan Memorial Hospital (No. 1090304). Upon completion of data collection, SPSS 22.0 statistical software was used for coding and documentation. Descriptive statistics and inferential statistics were performed based on the research objectives and variable characteristics.

## 3. Results

In this study, there were 68 male participants (42%) and 94 female participants (58%), with an average age of 53.2 ± 13.6 years old. A total of 80.9% were married, 71% had an education level above high school, and 63.9% reported a monthly income of NT 50,000–100,000 (USD 1550–3100). Additionally, 71% were employed full-time, and 96.9% resided in Changhua County, making up the majority. Furthermore, 158 people (97.5%) reported having religious beliefs. Regarding the relationship with the long-term care facility residents, 74.6% were either spouses or children, with 85.2% reporting a good relationship. On average, family members visited residents more than once a week, with a rate of 82.6%. Furthermore, 68.6% (111 people) of the residents had been hospitalized more than three times in the past six months, and for 85.2% of them, different family members were able to take turns caring for patients. Additionally, all family members had reserved a cooperation bed in the institution (Table 1).

### 3.1. Score of Stress and Coping Behavior

The average score on the stress scale was 65.62 with a standard deviation of 7.59, indicating that the overall stress experienced by family members of long-term care facility residents during hospitalization was at a medium to high level. Across the four dimensions of “physiology”, “psychology”, “economy”, and “life”, the scores ranked in the following order: “physiology”, “life”, “psychology”, and “economy”. This suggests that family members experienced higher stress related to “physiology” compared to “life”, “psychology”, and “economy”. On the coping behavior scale, the total mean score was 126.85 with a standard deviation of 16.91. Within the two dimensions of “emotional coping” and “problem coping”, the scores ranked in the following order: “problem coping” and “emotional coping”. This indicates that when faced with problems related to long-term care facility residents during hospitalization, more individuals choose problem-solving coping strategies. This section may be divided into subheadings and should provide a concise and precise description of the experimental results, their interpretation, and the conclusions drawn from the research (Table 2).

### 3.2. Inferential Statistics Analysis

#### 3.2.1. Difference between Personal Characters and Stress Subscale and Coping Behavior Scale

Further analysis revealed significant differences between the personal characteristics of family members of long-term care facility residents and the total stress score, particularly in terms of gender, taking turns caring for patients, and age. According to Scheffe’s post hoc comparative analysis, family members aged 39 or younger had significantly higher stress scores compared to those aged 40–64 (Table 3).

Analysis of the personal characteristics of family members of long-term care facility residents and the four dimensions of the stress scale revealed significant differences in age, education, economic status, work status, family relationship, and the relationship with the patient, as well as the stress scores. According to Scheffe’s post hoc comparative analysis, the psychological stress score of family members aged 39 or younger was significantly higher than those aged 40–64. Those with a junior school or senior high school education, and those with junior college or above, had higher physiology scores than those with primary school education or below. Those with primary school education or below had higher economy scores than those with junior school or senior high school education, and those with junior college or above. The participants with an economic status of NT 50–100 thousand, and those with more than NT 100 thousand, had higher physiology scores than those with an economic status of less than NT 50 thousand. Those with an economic status of more than NT 100 thousand had higher psychology scores than those with an economic status of less than NT 50 thousand, and those with an economic status of NT 50–100 thousand. The economy scores of participants with an economic status of less than NT 50 thousand, and those with an economic status of NT 50–100 thousand, were higher than those with an economic status of more than NT 100 thousand. Additionally, participants with an economic status of less than NT 50 thousand had higher economy scores than those with an economic status of NT 50–100 thousand. Participants with part-time and full-time work had higher physiology scores than those who did not work. The economic scores of participants with no work, and those with part-time work, were higher than those with full-time work. The participants whose family relationships were with their children had higher life scores than those whose family relationships were with others. Those with good family relationships had higher psychology scores than those with bad family relationships (Table 4).

Further analysis of the differences between the personal characteristics of the family members of long-term care facility residents and the two dimensions of “problem coping” and “emotional coping” revealed significant differences. The score of problem coping showed significant variations based on marital status, age, education, economic status, work status, and relationship. Additionally, there were significant differences in the score of emotional coping based on gender, marital status, family relationship, and admission during half of a year (Table 5).

According to Scheffe’s post hoc comparative analysis, the score of the problem coping for family members aged ≤39 and those aged 40–64 was significantly higher than for those aged ≥65. Individuals with an education level of junior school or senior high school, and junior college or above, exhibited higher problem coping scores compared to those with ≤primary school education. Subjects with an economic status of NT 50–100 thousand, and those with more than NT 100 thousand, had higher problem-coping scores than those with less than NT 50 thousand. Moreover, individuals with an economic status of more than NT 100 thousand had higher problem-coping scores than those with NT 50–100 thousand. Those employed full-time showed higher problem-coping scores than those without work. Family members who cared for had higher emotional coping scores than those caring for a spouse. Additionally, individuals with normal or good family relationships demonstrated higher problem-coping scores than those with poor family relationships. Finally, those who had been admitted during half of a year for ≥5 times had higher emotional coping scores than those admitted once or twice (Table 5).

#### 3.2.2. Correlation Analysis between Stress Subscale and Coping Behavior Subscale

Product-moment correlation analysis was employed to examine the correlation between stress and copying behavior. Table 6 showed that emotional coping was not correlated with life, and there was no correlation between problem coping, the total stress score, and its four sub-dimensions of physiology, psychology, economy, and life. However, emotional coping was positively correlated with the overall stress scale (R = 0.166, *p* < 0.05), physiology (R = 0.490, *p* < 0.001), and psychology (R = 0.505, *p* < 0.001), but negatively correlated with economy (R = −0.485, *p* < 0.001). Problem coping exhibited a positive correlation with life (R = 0.312, *p* < 0.001). The higher the stress scores for physiology and psychology, the more emotional coping strategies were chosen, whereas lower stress scores for the economy were associated with more emotional coping. Furthermore, higher stress scores in life were linked to greater utilization of problem-coping strategies (Table 6).

#### 3.2.3. Predictive Analysis between Significantly Factors of Coping Behaviors and Coping Behaviors

Multiple linear regression analysis was conducted to examine the relationship between stress and coping behaviors. Table 7 showed that the problem-oriented coping behaviors exhibited statistically significant associations with the overall constant (F = 24.278, *p* < 0.001), explaining 59.6% of the variance. It was found that physiology, psychology, income, work status, relationship, and problem-oriented coping behaviors significantly and positively predicted stress, whereas economy and problem-oriented coping behaviors significantly and negatively predicted stress. Furthermore, emotion-oriented coping behaviors demonstrated statistically significant associations with the overall constant (F = 8.548, *p* < 0.001), explaining 19.0% of the variance. Factors such as life, gender, admission within the past half year, and emotion-oriented coping behaviors significantly and positively predicted stress, while relationship and emotion-oriented coping behaviors significantly and negatively predicted stress (Table 8).

## 4. Discussion

The overall standardized average score for caregivers’ stress in this study is 57.03 points. The scores for the four major dimensions, from highest to lowest, are as follows: physiological dimension (3.79 points), lifestyle dimension (3.54 points), psychological dimension (3.45 points), and economic dimension (2.18 points). This indicates that the perceived stress among the caregivers in this study is moderately high. This result is lower than the average stress score of 3.7 reported by Chen et al. [21] for family members of long-term ventilator-dependent patients in five hospitals in Taipei. However, it is higher than the average stress score of 2.58 reported by Chen [22] for family members of end-of-life patients in palliative care units. This suggests that the stress experienced by family members of long-term institutionalized residents during hospitalization falls between that of family members caring for end-of-life patients and those caring for long-term ventilator-dependent patients. One possible explanation is that when patients are using ventilators for an extended period, it is often associated with severe or critical conditions, and family members not only have to care for the patient but also deal with various issues related to the ventilator, leading to increased anxiety and stress [23]. On the other hand, when long-term care facility residents are hospitalized, family members may have a lower level of familiarity with the resident’s health condition, and receiving a sudden notification of the hospitalization can result in an instantaneous increase in stress. This may contribute to higher stress levels compared to family members caring for end-of-life patients with chronic diseases.

In the stress scale’s sub-dimensions, the highest stress dimension in this study is the “physiological dimension (3.79 points)”, a result consistent with the findings of Kazemi et al. [24]. However, it differs from the study by Hsu Wen-juan et al. [25], which focused on primary caregivers of patients with cirrhosis at a medical center in northern Taiwan. In their research, the highest score was in the “lifestyle dimension (3.40 points)”. One possible explanation for this difference is that primary caregivers of patients with chronic illnesses often need to shift their entire focus to the elderly, including their past work, original lifestyle, and social activities. Time is not only reallocated, but most of it is spent with the elderly to fulfill the role of the primary caregiver. Hence, the stress on the lifestyle aspect is greater. According to a report by the Ministry of Health and Welfare, the average daily caregiving time for primary family caregivers in Taiwan is 11.06 h, significantly impacting personal lives [5]. Moreover, 20–22% of primary caregiver express dissatisfaction with their overall life situation [26,27]. Furthermore, family caregivers of chronic disease patients, after extended periods of caregiving, develop a profound understanding of the details involved in caring for older individuals. They seamlessly incorporate caregiving tasks into their daily lives, making them more adept at handling caregiving responsibilities and, consequently, experiencing lower levels of physical and physiological stress.

Conversely, the abrupt hospitalization of long-term care facility residents imposes additional and non-routine caregiving responsibilities on family members. The constant back and forth between the institution and the hospital adds to physical and life stress, which may be reflected in increased physiological stress. Approximately 46.45% of primary caregivers experience sleep disturbances when caring for disabled patients, and more than half (58%) of primary caregivers may suffer from insomnia or sleep-related issues. These factors contribute to increased physiological stress and fatigue, ultimately elevating the burden experienced by caregivers [26,28].

The study found that the lowest stress dimension is the “economic aspect”, which differs from Chen’s research on families of elderly dementia patients in Taipei City [29]. Chen’s study indicated that the primary caregivers scored highest on the “economic aspect”. This difference may be attributed to variations in patient characteristics. Caring for dementia patients involves endless and complex tasks, with caregivers predominantly comprising family members or friends who often provide unpaid or reduced-income caregiving. This can have an impact on the personal and family economic income of caregivers in dementia households. Unlike caregivers in long-term care facilities where residents’ family members may only be involved in short-term care during hospitalization, usually lasting 1–2 days or even a few hours, the economic pressure may not be immediately felt. Consequently, the focus of stress in these cases tends to be on physiological and psychological aspects. Additionally, regional economic factors may play a crucial role in influencing the perception of economic pressure. Chen’s study primarily focused on Taipei City [29], while this study centered around Changhua City. According to the 2022 statistical report from the Directorate General of Budget, Accounting, and Statistics, the average individual income in Taipei City is approximately NT 40,000, whereas, in Changhua City, it is around NT 32,000. However, the average monthly expenditure per person is approximately NT 29,000 in Taipei City and NT 15,800 in Changhua City [30]. Taking into account the mutual offset between income and expenditure, it can be observed that Changhua City has a higher average monthly surplus compared to Taipei City. This difference in financial dynamics may contribute to the lower perceived economic pressure in Changhua City. Moreover, in Changhua City, the roles of available social services, subsidies, and caregiver support programs include subsidies and allowances, community care centers, training and support programs, psychological counseling services, and so on. Furthermore, the occupational profile in Taipei City is characterized by a higher proportion of small self-employed businesses or company employers, which could lead to relatively higher economic incomes for some individuals, potentially surpassing the average personal income. This factor may contribute to a lower perception of economic pressure in this region.

In addition, Chen [29] indicated that the “economic aspect” is the primary burden for the main family caregiver, particularly in terms of “medical expenses” and “caregiver fees”. Tsai et al. [31] found a negative correlation between family monthly income and caregiver stress. When the monthly income of the family caregiver is higher, their perceived stress tends to be lower. Hu et al. [32] also highlighted a significant relationship between family income and the stress experienced by the primary caregiver, suggesting that caregivers may experience higher stress when the family income is lower. Furthermore, Zhang and Lu [26] pointed out that when there is a need for long-term care for older people in the family, 29.42% of households reported a deteriorating economic situation, with some facing a crisis. However, in Changhua, many long-term care facilities and hospitals have established contracts for “cooperation beds”. Cooperation beds operate on a 1:4 caregiver-to-patient ratio, meaning one caregiver takes care of four patients simultaneously. In other words, family members only need to bear one-fourth of the original caregiver fee. This unique mechanism significantly reduces the financial burden on families, alleviating the economic pressure that would have been present if they had opted for the traditional 1:1 caregiving model. During the period of this study, there was a severe outbreak of the COVID-19 pandemic, and the research institution was designated as a hospital responsible for managing COVID-19 cases. During this challenging pandemic period, primary caregivers may have experienced substantial stress related to hospital visits [33,34]. This could contribute to the relatively lower economic pressure experienced by primary caregivers compared to other stress dimensions.

In terms of the analysis comparing basic demographic data and perceived stress scores, this study found no significant differences in stress perception across various dimensions based on gender. This differs from many studies that suggest higher stress levels for female primary caregivers compared to males. Ruisoto et al. [34], in a study involving 688 primary caregivers of the elderly, found that female caregivers experienced greater stress than their male counterparts, with a significant difference. Similar findings have been reported by other scholars [14,35,36,37]. However, studies by Chen [38] and Shen et al. [39] align with the results of this research, suggesting that stress perception does not necessarily differ based on gender. The impact of gender on stress perception is an aspect that requires further exploration. It is also possible that the lack of significant differences in gender-related stress perception in this study could be influenced by the fact that the family caregivers typically hired professional caregivers to care for the patient during hospitalization, resulting in very brief periods of direct caregiving. As a result, gender may not manifest as a significant factor in stress perception across various dimensions. Nevertheless, additional research is needed to provide further support and validation for these findings.

This study found significant differences in stress perception based on economic factors among the participants. Specifically, caregivers with a family income below NT 50,000 experienced significantly greater economic pressure compared to those with incomes between NT 50,000 and 100,000 and those with incomes exceeding NT 100,000. This suggests that as economic conditions worsen, caregivers experience a heavier economic burden, aligning with the findings of many other studies. Chen [38] revealed that the average monthly income of a family is a crucial factor influencing economic burden. Those with incomes below NT 30,000 and incomes between NT 30,000 and 100,000 exhibited significantly higher economic stress compared to those with incomes exceeding NT 100,000. The lower the average monthly income of a family, the higher the economic stress burden on the caregiver. Similarly, a study by Hsu et al. [15] on caregivers of disabled older individuals in Macau indicated that when the family income was below HKD 10,000 (equivalent to approximately NT 43,350 at the time), the primary caregivers experienced significantly higher stress compared to those with incomes above HKD 10,000. Therefore, family income can be considered a critical predictor of caregiver stress. As the economic situation of caregivers improves, their perceived stress tends to decrease [40,41].

This study found significant differences in overall stress perception based on whether there was a rotation of family caregivers. Examining individual dimensions, it was observed that when the primary caregiver had no rotating family members to share the caregiving responsibilities, their physiological stress was significantly higher than those who had rotating family members involved. This finding is consistent with similar studies [14]. Research by Hsu [10] on caregivers of disabled older individuals in Macau noted that because caregivers often provide care to their own family members, the prolonged duration of caregiving can impact individuals across multiple dimensions, and additional caregiver support is necessary. Zhang and Lu [26] also mentioned that 51.39% of caregivers had no alternative caregivers available for rotation, resulting in significantly higher physiological stress compared to caregivers with rotating family members. In summary, the lack of rotating family caregivers may contribute to increased physiological stress for the primary caregiver, as observed in this study. These findings align with the notion that having additional caregiver support can help alleviate the stress experienced by the primary caregiver, especially when providing care for an extended period.

On the other hand, based on the two major dimensions of problem-focused and emotion-focused strategies, with an average score of 3.43 for problem-focused strategy and 3.02 for emotion-focused strategy, it can be inferred that family members of residents in long-term care facilities predominantly adopt problem-focused coping strategies. This finding is consistent with studies conducted by Alnazly [42] on 139 caregivers of hemodialysis patients and Kazemi et al. [24] on 110 caregivers of older stroke patients, both of which found that caregivers tend to choose problem-focused coping strategies when facing stress. However, Monteiro et al. [43], in a systematic review of 24 relevant studies, highlighted that family caregivers of dementia patients tend to prefer emotion-focused coping strategies when confronted with challenges. This result aligns with findings from other studies [44,45,46,47,48,49].

This study indicates that family members of residents in long-term care facilities predominantly choose problem-focused coping strategies, with an average score of 3.43, suggesting that their use of problem-focused coping falls between “sometimes” and “often”. This result is higher than that reported by Huang [50] in a study on coping behaviors of liver cancer patients before their first treatment. The research showed that liver cancer patients, upon initial diagnosis, also tended to choose problem-focused coping strategies, with an average score of 2.66, indicating that their use of problem-focused coping fell between “rarely” and “sometimes”. One possible reason for this difference could be that while both groups opt for problem-focused coping, patients facing a new diagnosis lack similar experiences in decision-making regarding their condition. Their frequency of selecting problem-solving methods may be lower initially. However, when residents of long-term care facilities are hospitalized, their family members have accumulated experience with hospital-related procedures and self-arrangement. This accumulated experience enables them to utilize previously chosen coping strategies when facing problems, resulting in a higher frequency of choosing problem-focused coping compared to liver cancer patients at the time of their initial diagnosis.

In this study, there was no significant difference in problem-focused coping behavior between genders among family members of residents in long-term care facilities. Problem-focused coping behaviors include developing a plan, seeking information, and creating a budget. Otherwise, emotion-focused coping behaviors include expressing emotions, practicing relaxation techniques, and joining support groups. The two coping behaviors have different implications for problem solving. This finding contrasts with several studies, indicating that male primary caregivers tend to adopt problem-focused coping behaviors more than female caregivers [24,51,52]. Kazemi et al. [24], in their study on 110 primary caregivers of older stroke patients, found that males were more likely to choose problem-focused coping strategies than females, with a significant difference observed. Similarly, Hassan et al. [51], in their investigation of 100 primary caregivers of patients with schizophrenia, noted that male caregivers tended to choose problem-focused coping behaviors, including reassessment of the problem and strategizing on how to deal with it, more frequently than female caregivers. Additionally, Alnazly [52], in their study on 139 primary caregivers of hemodialysis patients, reported that male caregivers were more likely to seek social support or access relevant resources when facing problems, indicating a greater inclination toward problem-focused coping behaviors compared to female caregivers. A possible explanation for the lack of significant gender differences in problem-focused coping behaviors among family members of long-term care facility residents in this study could be attributed to the standardized procedures for handling hospitalizations. Typically, emergency contact information provided by family members is used when residents need to be admitted to the hospital. Regardless of gender, individuals follow hospital instructions for hospitalization procedures, such as signing consent forms and agreeing to bed assignments. Consequently, the process of handling resident hospitalizations becomes routine, requiring minimal deviation in decision-making. Therefore, gender may not significantly influence problem-focused coping behaviors in the context of handling hospitalizations.

In this study, significant differences were observed in emotion-focused coping behaviors between genders among family members of residents in long-term care facilities. The research indicates that when facing problems, females tend to choose emotion-focused coping strategies significantly more than males, a finding consistent with several studies. Many females’ cultural and emotional interpretations are influenced by traditional values. They bear caregiving responsibilities in the family and are expected to demonstrate resilience and selflessness. This leads them to often handle stress in an emotion-focused manner when facing challenges. As society changes, more females are beginning to value self-expression and emotional health, fostering a new interpretation of female roles and more open emotional communication. For instance, Iavarone et al. [42] found in their study of 86 primary caregivers of Alzheimer’s disease patients that female caregivers were more likely than male caregivers to choose emotion-focused coping behaviors when dealing with problems, with a significant difference observed. Similarly, Bertolin et al. [44] reported in their research involving 107 chronic hemodialysis patients that females tended to choose emotion-focused coping behaviors, including avoidance, distancing, and confrontational emotional behaviors, with average scores higher than those of males. This gender difference in coping behaviors may be influenced by cultural and moral perceptions. Females are often viewed as the preferred choice for primary caregivers, and they may feel compelled to shoulder caregiving roles to meet societal expectations. However, this can lead to immense pressure on both physiological and psychological levels [53]. Even though support systems may be available, females often hesitate to seek assistance from others [54]. Consequently, females facing illness or significant stress may exhibit more emotional reactions, such as crying, anxiety, or avoidance, compared to males [42,44,45].

Furthermore, significant differences were found in emotion-focused coping behaviors based on the relationship to the resident, particularly between those who were children or daughters-in-law compared to other relationships. Considering these factors, when the primary caregiver during the resident’s hospitalization is a child or daughter-in-law, they may experience various pressures and a sense of duty, viewing caregiving for older people as an essential task for women. They may also be less inclined to seek external resources. Under such immense pressure, females may internalize stress into emotional responses, potentially leading to negative emotions. Therefore, females may exhibit a higher tendency towards emotion-focused coping behaviors when facing problems as compared to males.

In this study, significant differences were observed in problem-focused coping behaviors concerning the age, education level, family income, and employment status of the participants. Young and middle-aged adults scored higher in problem-focused coping behaviors compared to older individuals. This could be attributed to the fact that as individuals age, they may develop health problems and become apprehensive about the stress associated with illness. Additionally, some older individuals may exhibit more rigid behavior patterns and have difficulty maintaining social interaction skills, leading to an increased probability of cognitive impairments and lower acceptance of change, consequently affecting their problem-solving abilities [55]. Participants with higher levels of education scored higher in problem-focused coping behaviors compared to those with lower levels of education. This may be because individuals with higher education have a broader and more diverse learning experience, resulting in significant differences in their coping abilities [56,57]. Regarding family income, participants from higher-income families exhibited higher levels of problem-focused coping behaviors compared to those from lower-income families. This could be attributed to the greater availability of resources among wealthier families [58]. Concerning employment status, full-time employees showed higher levels of problem-focused coping behaviors compared to unemployed individuals. This may be because primary caregivers who are full-time employees need to efficiently solve immediate problems within limited time constraints, making problem-focused coping their preferred choice.

This study demonstrates a positive correlation between total family caregiver stress scores and problem-focused coping behaviors, consistent with the findings of Arif et al. [59] and Bawalsah [60]. However, it differs from the results of Siciliano et al. [61], Lee et al. [62], Lan et al. [63], and Kazemi et al. [24], which indicate a positive correlation between total family caregiver stress scores and emotion-focused strategies among caregivers of children with illnesses. One possible explanation is that when the primary caregiver is caring for a child, they may experience heightened emotional distress due to the child’s young age, leading to feelings of sadness, inability to cope, and possibly self-blame, resulting in more emotional responses. Hence, their total stress scores are positively correlated with emotion-focused coping behaviors. Furthermore, the study found a negative correlation between caregiver stress and problem-focused coping; as total stress scores increase, primary caregivers are less likely to choose problem-focused strategies [56]. Additionally, studies by Lloyd et al. [64] and Yuan et al. [65] on primary caregivers of dementia patients also found a correlation between total stress scores and emotion-focused coping behaviors, but it was negative. That is, when total stress scores are lower, more primary caregivers tend to choose emotion-focused coping behaviors. These results warrant further investigation.

In addition, this study shows a positive correlation between the psychological dimension of caregiver stress and problem-focused coping, consistent with the findings of Monteiro [43] regarding the primary caregivers of dementia patients. However, this study found no correlation between psychological stress and emotion-focused coping behaviors, which differs from the findings of several scholars [47,48,62]. Lee et al. [62] studied primary caregivers of children with Down syndrome, while other studies focused on the caregivers of Alzheimer’s patients. Whether the difference in the type of illness cared for by the participants contributes to this disparity deserves further investigation. On the other hand, since the residents in this study had multiple hospitalizations in the past six months, family members may prioritize solving the immediate and practical problem of “hospitalization”. Therefore, even with higher levels of psychological stress, family members may still choose problem-focused strategies as their primary approach. Consequently, it can be reasonably inferred from this why the results of this study show a positive correlation between psychological stress and problem-focused coping behaviors but no correlation with emotion-focused coping behaviors.

Furthermore, this study demonstrates a negative correlation between the economic dimension and problem-focused coping, with no correlation with emotion-focused coping. This finding differs from that of Frota et al. [45], who studied 70 family caregivers in intensive care units and found that greater economic stress among primary caregivers was correlated with emotion-focused coping behaviors. Additionally, Chan and Wong [65] showed that caregivers facing greater financial pressure may tend to rely more on emotion-focused strategies, indicating a positive correlation between the two. The reason for the different results of this study and others may be attributed to the circumstances during the emergency department visit, where the hospital only charges family members for the emergency registration fee, while other expenses such as room charges, nursing fees, and out-of-pocket expenses are settled upon discharge. Therefore, family members may not have a clear understanding of or immediate exposure to the associated costs, resulting in lower economic stress in the short term. Moreover, the participants in this study were mostly the emergency contacts of the patients, and their primary goal upon arriving at the emergency department was to address the patient’s acute issues and cooperate with the hospital for admission-related matters. Hence, problem-focused coping behaviors were predominantly observed.

In terms of predictor variables, this study demonstrates that the physiological dimension, psychological dimension, economic dimension, family income, employment status, and relationship quality with the resident have significant predictive capabilities for problem-focused coping behaviors. Conversely, the lifestyle dimension, gender, marital status, and number of hospitalizations in the past six months have significant predictive capabilities for emotion-focused coping behaviors. These findings are consistent with Kazemi et al. [24], who suggested that physiological stress can serve as a predictor for problem-focused coping behaviors. Additionally, Roberts et al. [66] and Roberts and Ishler [67] found that individuals with better relationships with family members or long-term care facility residents are more likely to choose problem-focused coping behaviors. The relationship quality with the resident being a predictive factor for selecting problem-focused coping behaviors is consistent with the findings of this study. Moreover, Coppetti et al. [56] demonstrated a positive correlation between education level and problem-focused coping behaviors, which supports the results of this study. Additionally, Wang et al. [23] indicated that education level and economic status can serve as predictors for using problem-focused coping behaviors. However, in this study, education level did not exhibit significance as a predictor for problem-focused coping behaviors.

Regarding the predictive factors for emotion-focused coping behaviors, various studies have shown different results. For instance, studies examining family income [45], gender [42,45], psychological stress [47,68], and emotional reactions have yielded findings that differ from those of this study. However, there is no consistent conclusion among these studies. It is worth exploring further whether regional cultural differences, characteristics of the care recipient, or the duration of caregiving contribute to these discrepancies.

## 5. Conclusions

This study focused on family caregivers of residents in a long-term care facility within a teaching hospital in central Taiwan. Its aim was to explore the correlation between residents’ hospitalization and caregivers’ stress and coping behaviors. Based on the research objectives, the conclusions are as follows:(1)The primary caregivers of hospitalized patients in long-term care facilities are predominantly children, daughters-in-law, and sons-in-law, accounting for 70.9%. Female caregivers make up 58.0%, with most caregivers being middle-aged, and 80.9% are married. The overall family income is mostly between NT 50,000 and NT 100,000, which is considered middle class. A majority (83.3%) of caregivers are still employed, and 77.2% visit the facility at least once a week. During the hospitalization of residents, 85.2% of primary caregivers have others to help share caregiving responsibilities.(2)The standardized average total stress level among primary caregivers is moderate. Stress is highest in the physiological domain, followed by lifestyle, with the lowest in the economic domain. Caregivers in this study primarily used problem-focused coping behaviors rather than emotion-focused coping behaviors.(3)There is a correlation between total stress scores and stress in the physiological, psychological, and economic domains with problem-focused coping behaviors. Additionally, stress in the lifestyle domain is associated with emotion-focused coping behaviors.(4)Factors such as physical, psychological, and economic stress, family income, employment status, and relationship with the resident significantly predict problem-focused coping behaviors. Lifestyle changes in the past six months, gender, marital status, and hospitalization frequency significantly predict emotion-focused coping behaviors.

Based on the above key research findings, this study highlights that the main stress faced by family caregivers of long-term care facility residents during hospitalization is physiological stress. Those without a substitute caregiver and those under the age of 39 tended to experience higher overall stress scores. It is recommended that clinical medical staff should proactively care for the physical health of caregivers during the hospitalization of long-term care facility residents to reduce their stress and burden. In addition, family caregivers of long-term care facility residents often lack basic elderly care concepts and skills due to their reliance on the institutional care model. Therefore, when the elderly are hospitalized, nursing staff should strengthen the training of family caregivers in caring knowledge and skills. Healthcare facilities are also encouraged to establish support groups for primary caregivers of long-term care facility residents. When a resident is hospitalized, departments within these support groups can proactively send hospital volunteers or social workers to assist and support primary caregivers.

Taiwan’s long-term care 2.0 (LTC 2.0) policy focuses on subsidies for home care and daycare services, and the subsidies for institutional residents are relatively insufficient. These caregivers can only enjoy a specific tax deduction of NT 120,000 per year, and these regulations do not apply uniformly to different income groups. It is recommended that future policies consider including caregivers of long-term care facility residents within the scope of LTC 2.0 services in order to make more comprehensive use of government resources to meet the care needs of the people.

Due to limitations in time, manpower, epidemic situation, and financial resources, this study was unable to issue a large number of questionnaires to fully qualified participants for a comprehensive survey. This limitation may lead to bias in the randomness of the sample selection. In addition, this study only selected participants from teaching hospitals in the Changhua City area, which may cause the results to be different from respondents in other counties and cities. Therefore, the findings may not be generalizable to other groups due to the characteristics of the sample.

Furthermore, during this study, which coincided with the COVID-19 pandemic, whether primary caregivers experienced different caregiving stress and coping behaviors due to special circumstances can be further investigated in future studies using longitudinal research methods or combining qualitative interview data. It is also recommended that future studies include the lived experiences of primary caregivers while assessing their physical and mental health, stress levels, and coping strategies. These additions may enhance the completeness and power of the study results.

## Figures and Tables

**Table 1 healthcare-12-02022-t001:** Personal characteristics (n = 162).

	Variables	n	%
Gender	Male	68	42
	Female	94	58
Age	M ± SD	53.2 ± 13.6	
	≤39	30	18.5
	40–64	100	61.7
	≥65	32	19.8
Marital status	Married	131	80.9
	Unmarried/widowed/divorced	31	19.1
Education	No formal education	29	17.9
	Primary school	18	11.1
	Junior high school or above	115	71
Income	NT < 50 thousand	46	28.4
	NT 50–100 thousand	103	63.9
	NT > 100 thousand	13	8.0
Work status	None or part-time	47	29
	Full-time	115	71
Lived county	Changhua County	157	96.9
	Others	5	3.1
Religion	No	4	2.5
	Yes	158	97.5
Family relationship	Spouse or children	121	74.6
	Others	41	25.4
Relationship	Bad	24	14.8
	Normal or above	138	85.2
Visiting frequency	Once a week or above	133	82.6
	Less than once a week	29	17.4
Admission during half of a year	Three times or above	111	68.6
	Less than three times	51	31.4
Take turns	No	24	14.8
	Yes	138	85.2

**Table 2 healthcare-12-02022-t002:** Score of stress and coping behavior.

Items	Range	Sum/Mean	SD
Overall stress scale	20–100	65.62 (3.28)	7.59
Physiology	3–15	11.37 (3.79)	2.13
Psychology	6–30	20.73 (3.45)	3.69
Economy	4–20	8.72 (2.18)	3.49
Life	7–35	24.78 (3.54)	3.75
Overall coping behavior scale	40–200	126.85 (3.17)	16.91
Problem coping	15–75	51.47 (3.43)	10.23
Emotional coping	25–125	75.38 (3.02)	7.83

**Table 3 healthcare-12-02022-t003:** Analysis between personal characteristics of the family members of long-term care facility residents and the total score of stress n = 162.

Variables		n	Overall Stress Scale			
			M	SD	t/F	*p*
Gender	Male	68	58.90	6.82	−2.29	0.02 *
	Female	94	56.29	6.36		
Marital Status	Unmarried/widowed/divorced	31	55.45	6.68	0.16	0.88
	Married	131	55.24	6.66		
Take turns caring for patients	No	24	59.08	9.02	−3.12	0.002 **
	Yes	138	54.62	5.93		
Age	① ≤39	30	58.77	10.09	6.07	0.003 ** ① > ②
	② 40–64	100	54.12	5.41		
	③ ≥65	32	56.66	4.93		
Education	① ≤Primary school	29	55.03	10.6	0.95	0.39
	② Junior school or senior high school	38	56.58	5.30		
	③ Junior college or above	95	54.84	5.47		
Economic status	① NT < 50 thousand	46	55.96	7.67	0.69	0.50
	② NT 50–100 thousand	103	55.20	6.30		
	③ NT > 100 thousand	13	53.54	5.44		
Work status	① None	27	53.78	8.63	2.86	0.06
	② Part-time	20	58.30	7.51		
	③ Full-time	115	55.11	5.81		
Lived county	① Changhua County	68	55.60	6.22	0.52	0.61
	② Others	94	55.05	6.95		
Religion	① None	4	58.25	9.91	0.46	0.63
	② Taoism/Buddhism	141	55.15	6.80		
	③ Christianity or Catholicism	17	55.71	4.31		
Family relationship	① Spouse	26	55.46	8.98	1.38	0.25
	② Children	115	55.65	6.18		
	③ Others	21	53.05	5.55		
Relationship	① Bad	24	55.04	5.58	0.19	0.83
	② Normal	53	54.91	7.32		
	③ Good	85	55.59	6.54		
Visiting frequency	① Once a month or above	133	55.27	6.75	−0.54	0.96
	② Less than once a month	29	55.34	6.36		
Admission during half of a year	① Once or twice	51	55.12	5.91	0.03	0.97
	② Three–four times	55	55.29	5.79		
	③ ≥Five times	56	55.43	8.03		

* *p* < 0.05, ** *p* < 0.01.

**Table 4 healthcare-12-02022-t004:** Analysis of personal characteristics of the family members of long-term care facility residents and the scale of stress n = 162.

Item	n	Physiology		Psychology		Economy		Life	
		M	t/F	M	t/F	M	t/F	M	t/F
Gender									
Male	68	11.13	−1.18	16.54	−2.81	8.63	−0.37	17.59	−0.99
Female	94	11.53		17.93		8.84		17.99	
Marital status									
Unmarried/widowed/divorced	31	10.84	−1.53	16.45	−1.77	10.45	3.09	10.45	−0.27
Married	131	11.49		17.56		8.35	**	8.35	
Take turns									
No	24	11.09	−4.04	17.26	−0.82	8.29	−4.27	17.98	1.92
Yes	138	12.92		17.83		11.42	**	16.92	
Age									
≤39	30	11.97	3.05	18.73	4.90	9.40	3.12	18.67	2.18
40–64	100	11.41		16.80		8.23	**	17.68	
≥65	32	10.66		17.75		9.78		17.47	
Education									
① ≤Primary school	29	9.79	10.73	6.10	2.85	11.79	19.07	17.34	1.27
② Junior or Senior high school	38	11.68	***	17.50		9.08	***	18.32	
③ Junior college or above	95	11.72	② > ① ③ > ①	17.66		7.69	① > ② ① > ③	17.77	
Income									
① NT < 50 thousand	46	10.39	8.38	16.57	5.30	11.50	37.57	7.50	2.14
② NT 50–100 thousand	103	11.65	***	17.40	**	8.06	***	18.10	
③ NT > 100 thousand	13	12.54	② > ① ③ > ①	19.69	③ > ① ③ > ②	4.54	① > ② ① > ③ ② > ③	16.77	
Work status									
① None	27	9.56	15.56	16.19	2.80	11.30	18.86	16.74	3.04
② Part-time	20	12.50	***	16.90		10.90	***	18.00	
③ Full-time	115	11.59	② > ① ③ > ①	17.70		7.78	① > ③ ② > ③	18.04	
Lived county									
① Changhua	68	11.29	−0.35	17.88	1.86	8.57	−0.56	17.85	0.14
② Others	94	11.41		16.96		8.88		17.80	
Family relationship									
① Spouse	26	10.88	1.62	16.65	1.16	10.19	2.78	17.73	3.75
② Children	115	11.56		17.58		8.43		18.09	*
③ Others	21	10.90		16.90		8.76		16.48	② > ③
Relationship									
① Bad	24	11.38	0.01	15.88	4.75	9.58	0.92	18.21	0.33
② Normal	53	11.34		17.02	*	8.79		17.75	
③ Good	85	11.38		17.96	③ > ①	8.49		17.75	
Visiting frequency									
① Once a month or above	133	11.32	−0.62	17.45	0.91	8.57	−1.43	17.93	1.20
② Less than once a month	29	11.59		16.86		9.59		17.31	
Admission during half of a year									
① Once or twice	51	11.57	1.34	17.16	0.18	8.80	0.03	17.59	0.30
② Three–four times	55	10.98		17.53		8.80		17.98	
③ ≥Five times	56	11.55		17.34		8.66		17.88	

* *p* < 0.05, ** *p* < 0.01, *** *p* < 0.001.

**Table 5 healthcare-12-02022-t005:** Analysis between personal characteristics of the family members of long-term care facility residents and the scale of coping behavior n = 162.

Item	n	Problem Coping			Emotional Coping		
		M	t/F	*p*	M	t/F	*p*
Gender							
Male	68	51.04	−0.45	0.65	73.50	−2.65	0.01**
Female	94	51.78			76.74		
Marital status							
Unmarried/widowed/divorced	31	48.06	−2.08	0.04 *	78.42	2.44	0.02*
Married	131	52.27			74.66		
Take turns							
No	24	51.92	−0.23	0.82	75.08	0.20	0.84
Yes	138	51.39			75.43		
Age							
① ≤39	30	54.23	4.90	0.01 **	76.63	0.48	0.62
② 40–64	100	52.14		① > ③	75.15		
③ ≥65	32	46.78		② > ③	74.94		
Education							
① ≤Primary school	29	43.45	13.36	0.000 ***	73.86	0.81	0.45
② Junior or senior high school	38	51.53		② > ①	75.13		
③ Junior college or above	95	53.89		③ > ①	75.95		
Income				0.000 ***			
① NT < 50 thousand	46	45.41	30.55	② > ①	75.87	0.42	0.66
② NT 50–100 thousand	103	52.28		③ > ①	75.39		
③ NT > 100 thousand	13	66.46		③ > ②	73.62		
Work status							
① None	27	46.48	4.00	0.02 *	73.56	2.37	0.09
② Part-time	20	52.75		③ > ①	73.00		
③ Full-time	115	52.42			76.23		
Lived County							
① Changhua	68	51.54	0.88	0.94	75.03	−0.49	0.63
② Others	94	51.41			75.64		
Religion							
① None	4	59.25	1.26	0.29	77.75	2.55	0.08
② Taoism/Buddhism	141	51.38			74.86		
③ Christianity/Catholicism	17	50.41			79.18		
Family relationship							
① Spouse	26	49.31	0.80	0.45	72.19	3.44	0.03 *
② Children	115	52.06			76.35		② > ①
③ Others	21	50.90			74.05		
Relationship							
① Bad	24	44.08	9.52	0.00 ***	77.96	1.84	0.16
② Normal	53	51.00		② > ①	74.28		
③ Good	85	53.85		③ > ①	75.34		
Visiting frequency							
① Once a month or above	133	51.72	0.67	0.50	75.65	0.92	0.36
② Less than once a month	29	50.31			74.17		
Admission during half of a year							
① Once or twice	51	51.80	1.91	0.34	72.88	3.96	0.02 *
② Three–four times	55	49.89			76.42		③ > ①
③ ≥Five times	56	52.71			76.64		

* *p* < 0.05, ** *p* < 0.01, *** *p* < 0.001.

**Table 6 healthcare-12-02022-t006:** The relationship between the stress and coping behavior n = 162.

Iten	Emotional Coping		Problem Coping	
	r	*p*	r	*p*
Overall stress scale	0.166	0.035 *	0.144	0.067
Physiology	0.490	0.000 ***	−0.091	0.252
Psychology	0.505	0.000 ***	0.065	0.412
Economy	−0.485	0.000 ***	0.046	0.562
Life	0.060	0.445	0.312 ***	0.000 ***

* *p* < 0.05, *** *p* < 0.001.

**Table 7 healthcare-12-02022-t007:** The multiple linear regression for stress and problem-oriented coping behaviors n = 162.

Item	Overall Scale Scores						
	B	β	T	*p*	VIF		
Overall constant	25.278		3.709	0.000 ***	1.181~8.215		
Overall stress	0.009	0.006	0.039	0.969			
Physiology	1.711	0.357	4.677	0.000 ***			
Psychology	0.831	0.256	2.468	0.015 *			
Economy	−1.227	−0.418	−3.972	0.000 ***			
Marital status	−2.433	−0.094	−1.654	0.100			
Age	0.210	0.013	0.175	0.861			
Education	1.077	0.082	1.099	0.273			
Income	3.073	0.171	2.400	0.018 *			
Work status	−3.090	−0.231	−3.063	0.003 **			
Relationship	2.663	0.190	3.493	0.001 **			

Note 1: * *p* <0.05; ** *p* < 0.01; *** *p* < 0.001.

**Table 8 healthcare-12-02022-t008:** The multiple linear regression for stress and emotion-oriented coping behaviors n = 162.

Item	Overall Scale Scores						
	B	β	T	*p*	VIF		
Overall constant	54.989	5.898	9.324	0.000 ***	1.007~1.078		
Life	0.932	0.301	4.197	0.000 ***			
Gender	3.197	0.202	2.833	0.005 **			
Marital status	−3.924	−0.198	−2.706	0.008 **			
Family relationship	1.112	0.077	1.041	0.300			
Admission during half of a year	1.790	0.186	2.617	0.010 *			

Note 1: * *p* < 0.05; ** *p* < 0.01; *** *p* < 0.001.

## Data Availability

The data presented in this study are available on request from the corresponding author. The data are not publicly available to retain participant privacy.

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
