# Peer review of "Stress and Coping Behavior Exhibited by Family Members Toward Long-Term Care Facility Residents While Hospitalized"

_healthcare, 2024, doi:10.3390/healthcare12202022_

Round 1
Reviewer 1 Report
Comments and Suggestions for Authors
Han-Lin Kuo and Yi-Wen Chiu submitted the paper titled Stress and coping Behavior of Family Members for Long-term Care Facility Residents in Hospitalization.
The paper is clear, nicely written and understandable.
My comments are below:
The study effectively presents the differences in stress dimensions (physiological, lifestyle, psychological, and economic) among caregivers of long-term care facility residents during hospitalization. The comparison with existing literature provides valuable context and enhances the validity of the findings. However, it would be beneficial to explore the reasons behind the discrepancies in stress dimensions further, such as the impact of regional healthcare policies or support systems on caregivers' stress levels.
The study reports that the economic dimension of stress is the lowest among caregivers in this setting. While potential reasons are discussed, such as differences in income and cost dynamics between Taipei City and Changhua City, the explanation would benefit from more in-depth analysis. For example, examining the role of social services, subsidies, or caregiver support programs available in Changhua could provide more insights into why economic stress is perceived to be lower.
The study's findings on the lack of significant gender differences in stress perception but significant differences in emotion-focused coping behaviors align with some literature but contradict others. Further discussion on the cultural and social norms affecting gender roles in caregiving in the region, and how they might impact stress and coping behaviors, could add depth to the interpretation of these results.
While the study highlights the relationship between family income and perceived economic stress, it would be helpful to incorporate additional variables, such as employment status, access to financial support, or caregiving costs, that may also influence economic stress. A more comprehensive analysis could improve the understanding of economic burdens faced by caregivers, especially in regions with different socio-economic conditions.
Comments on the Quality of English LanguageMinor English editing is needed.
Author Response
Dear Reviewer 1,
Thank you to the editors and reviewers for taking the time to review our manuscript.
We also appreciate for your valuable comments and advice. These comments are all valuable and very helpful for revising and improving our paper. We have revised the manuscript accordingly, and the detailed point-by-point responses are listed below. Additionally, we have provided explanations and supplements addressing each of the comments.
Reviewer1:
The study reports that the economic dimension of stress is the lowest among caregivers in this setting. While potential reasons are discussed, such as differences in income and cost dynamics between Taipei City and Changhua City, the explanation would benefit from more in-depth analysis. For example, examining the role of social services, subsidies, or caregiver support programs available in Changhua could provide more insights into why economic stress is perceived to be lower.
The study's findings on the lack of significant gender differences in stress perception but significant differences in emotion-focused coping behaviors align with some literature but contradict others.
Further discussion on the cultural and social norms affecting gender roles in caregiving in the region, and how they might impact stress and coping behaviors, could add depth to the interpretation of these results.
While the study highlights the relationship between family income and perceived economic stress, it would be helpful to incorporate additional variables, such as employment status, access to financial support, or caregiving costs, that may also influence economic stress. A more comprehensive analysis could improve the understanding of economic burdens faced by caregivers, especially in regions with different socio-economic conditions.
Ans:
(1) Thank you for your kind comments on our manuscript. We have included additional more in-depth analysis in the manuscript, as outlined on page 12, lines 347-350.
(2)Thank you for your kind comments on our manuscript. We have included additional discussion on the cultural and social norms affecting gender roles in caregiving in the region in the manuscript, as outlined on page 15, lines 478-483.
(3) Thank you for your kind comments on our manuscript. We have included additional more in-depth analysis in the manuscript, as outlined on page 12, lines 347-350.

Reviewer 2 Report
Comments and Suggestions for Authors
Authors
This manuscript is challenging to follow due to both the command of the English language, overall organization, including research design, methods, findings and discussion. This manuscript may be better received as two separate manuscripts. It appears that the authors created an instrument to measure stress, without fully discussing how the instrument was developed and how subscales were assigned, there is no mention of factor analysis. The findings discuss levels without cutoffs and scoring. Cronbachs are provided for the instruments however the Cronbach discussion lacks clarity regarding the study that manuscript is reporting on. The analysis is extensive however the analysis of the findings appear to be incomplete in regards to the research questions. The many tables lengthen the paper and it is not clear how they augment the paper. The discussion seems to focus heavily on what is known without integration of the findings into the discussion. What is known is presented study by study without integration of the multiple studies that are reviewed. An in text citation is needed on page 18, lines 627-629.
Comments on the Quality of English LanguageThis manuscript is challenging to follow due to both the command of the English language, overall organization, including research design, methods, findings and discussion.
Author Response
Dear Reviewer 2,
Thank you to the editors and reviewers for taking the time to review our manuscript.
We also appreciate for your valuable comments and advice. These comments are all valuable and very helpful for revising and improving our paper. We have revised the manuscript accordingly, and the detailed point-by-point responses are listed below. Additionally, we have provided explanations and supplements addressing each of the comments.
Reviewer2:
This manuscript is challenging to follow due to both the command of the English language, overall organization, including research design, methods, findings and discussion. This manuscript may be better received as two separate manuscripts.
It appears that the authors created an instrument to measure stress, without fully discussing how the instrument was developed and how subscales were assigned, there is no mention of factor analysis.
The findings discuss levels without cutoffs and scoring. Cronbachs are provided for the instruments however the Cronbach discussion lacks clarity regarding the study that manuscript is reporting on.
The analysis is extensive however the analysis of the findings appear to be incomplete in regards to the research questions. The many tables lengthen the paper and it is not clear how they augment the paper. The discussion seems to focus heavily on what is known without integration of the findings into the discussion.
What is known is presented study by study without integration of the multiple studies that are reviewed. An in text citation is needed on page 18, lines 627-629.
Ans:
- Thank you for your suggestion. To fully present the entirety of this study, the researcher has decided to use a single manuscript to convey the entire research process. As for the parts where the research design was not clearly articulated, we have rewritten and clarified those sections. (Line 107-146)
- The questionnaire for this study was developed by the researcher based on a review of numerous academic sources, and its content validity was assessed by five experts in the field. The stress subscale was a self-developed instrument structured around four dimensions: “physiology”, “psychology”, “economy”, and “life”, comprising a total of 20 questions. The Likert scale utilizing a five-point scoring method was adopted (total score: 20-100), with higher scores indicating greater stress experienced by the family members For further details, refer to lines 121-141.
- The analysis of the results in this study was conducted based on the key research objectives. The researcher has reviewed the findings in light of the reviewers' suggestions and discussed the research results accordingly. Thank you for your valuable feedback.
- The conclusion section of lines 627-629 in the original text summarized the key findings of this study. We have since revised it to provide a more concise statement (Line 603-623). Thank you for your feedback.

Reviewer 3 Report
Comments and Suggestions for Authors
Thank you for this paper, and I enjoyed reading it.
You started with a clear explanation of the current situation in Taiwan, and you have highlighted a very important issue.
There are some areas where the work would benefit from further clarification, and I have identified a few areas below. A paragraph on the context of health care delivery in hospitals in Taiwan, would be very useful, and would address a number of my comments below.
I wish you success in your ongoing research, and I hope you can build on this work.
In the introduction, the overall argument around an ageing population and the consequential increase in care homes. What is not explained, is why additional care-giving is required when the older person is hospitalised. Please add in a few sentences to set the context of care in the home versus care in the hospital. This may also include an explanation of what a cooperation bed is (line 165), as I do not know what you mean, and clarity would help an international audience.
Lines 52 and 56: I am not sure what you mean by functionality of families. Please explain.
Clarify that the cross-sectional survey (a quantitative method) was employed by administering the questionnaire in a face-to-face environment. When you say that you conducted face-to-face interviews, this implies a qualitative study.
The methods section needs to include information about how you recruited and selected the study sample. There also should be some detail regarding the characteristics or content of each of the domains. For example, what was included in the physiology domain? It is important that the content of each domain is presented, perhaps as a table or an appendix. This would also help in the results section as it is unclear what better physiology means, and it is also unclear what higher economy scores (line 203) mean. Is a high score having a better income, or greater hardship? Is ‘life’, a validated quality of life measure?
In addition, it would be useful to give some examples of what you mean by problem solving coping strategies and emotional coping strategies. This would also help inform the discussion.
The validity of the two scales (the self-developed scale and the Lin-based scale) was well described.
In section 3.1. you present the four dimensions, and then state the order in which they were ranked, but this did not fit with the sentence beginning in line 172, where the words seem to refer to the first set of dimensions. In line 181, remove the word experiment, as this study was not an experiment.
Table 2: clarify the meaning of the bracketed numbers in the ‘Mean’ column. Later on, in the discussion, these numbers represent the points scored for the domains, but I could not see how these points were derived.
When using Likert scales, it is often better to use medians and interquartile rankings. Doing so, would give a better sense of where the scores sit, i.e. whether in the low, medium or high bands of each domain. There may be very good reasons why you chose to use Means, and if so, please explain.
In section 3.2 you have highlighted the statistically significant differences, and it would be useful to include the direction of this difference within the text. For example there was a statistically significant difference between men and women, with men being more stressed.
In section 3.2.3 you refer to results that positively predicted stress and other behaviours and negatively predicted stress. It may be useful to clarify or re-word ‘negatively predicted’.
The discussion raises some interesting points, however, it is unlikely that with a sample size of n = 168, that some of the assumptions can be generalized, especially as there was no mention of a power calculation having been carried out.
Later in the discussion (lines 361 on) you refer to a 1:4 caregiver to patient ratio, and a few is also mentioned. This needs to be clarified for the international readership. What this sentence suggests, is that a care-giver (a family member of one patient), looks after four patients (their own family member plus three others), whilst the patients are in hospital, and they pay a fee to do this. Is this the case? If so, this is not a model of care that I (or others) will be familiar with, and thus, it would need to be clarified and explained. Later on, you refer to professional care-givers. Who are these, and where are the healthcare staff (nurses doctors etc.)?
When you refer to ‘sometimes’, ‘often’ etc. (lines 431 onwards), this does not really give an indication of the meaning. As mentioned above, there needs to be an example of the context of these domains. What were your participants asked? This section also refers to patients’ coping, rather than care-givers, as in your study, and I don’t think that this is the same thing.
The discussion would benefit from being written more succinctly. For example, the section from line 470, is repetitive in places.
The conclusion brings in a number of previously undiscussed concepts, including speculation and recommendations around nursing staff, and their roles (e.g. line 629). There does not seem to be evidence for many of these claims, and so I suggest that the conclusions are kept very short, and reflect the size and type of this study.
The limitations were reasonable, as were the recommendations.
Please proof-read the work for English language
Editing checks:
Lines 12 and 13: check for clarity
Line 18: Please do not use the word ‘object’ for people. I suggest you use participant or another more person-centered term
Line 27: correct the spelling of significant and behaviours
Line 38: I am not sure that reach is the correct word.
Lines 48+ Please use the future tense, where applicable
Line 67: Please use more person-centered words. Demented individuals is not person-centered. Perhaps older people living with disabilities and/or dementia.
Line 75: Do you mean papers? Perhaps studies might work as well?
Line 152: Performed
Lines 155, 212+: Please use participants rather than subjects.
Line 157: Include the unit of currency, and it might be useful to add a USD equivalent, or a reference to the mean income, for context?
Line 272: include a space before 'the'
Comments on the Quality of English LanguageAlthough the sentence structure was generally good, there were some errors (highlighted above). The would be benefit in trying to write more succinctly. This would improve the clarity of the work.
Author Response
Dear Reviewer 3,
Thank you to the editors and reviewers for taking the time to review our manuscript.
We also appreciate for your valuable comments and advice. These comments are all valuable and very helpful for revising and improving our paper. We have revised the manuscript accordingly, and the detailed point-by-point responses are listed below. Additionally, we have provided explanations and supplements addressing each of the comments.
Reviewer3:
You started with a clear explanation of the current situation in Taiwan, and you have highlighted a very important issue.
There are some areas where the work would benefit from further clarification, and I have identified a few areas below. A paragraph on the context of health care delivery in hospitals in Taiwan, would be very useful, and would address a number of my comments below.
I wish you success in your ongoing research, and I hope you can build on this work.
In the introduction, the overall argument around an ageing population and the consequential increase in care homes. What is not explained, is why additional care-giving is required when the older person is hospitalised. Please add in a few sentences to set the context of care in the home versus care in the hospital. This may also include an explanation of what a cooperation bed is (line 165), as I do not know what you mean, and clarity would help an international audience.
Lines 52 and 56: I am not sure what you mean by functionality of families. Please explain.
Clarify that the cross-sectional survey (a quantitative method) was employed by administering the questionnaire in a face-to-face environment. When you say that you conducted face-to-face interviews, this implies a qualitative study.
The methods section needs to include information about how you recruited and selected the study sample. There also should be some detail regarding the characteristics or content of each of the domains. For example, what was included in the physiology domain? It is important that the content of each domain is presented, perhaps as a table or an appendix. This would also help in the results section as it is unclear what better physiology means, and it is also unclear what higher economy scores (line 203) mean. Is a high score having a better income, or greater hardship? Is ‘life’, a validated quality of life measure?
In addition, it would be useful to give some examples of what you mean by problem solving coping strategies and emotional coping strategies. This would also help inform the discussion.
The validity of the two scales (the self-developed scale and the Lin-based scale) was well described.
In section 3.1. you present the four dimensions, and then state the order in which they were ranked, but this did not fit with the sentence beginning in line 172, where the words seem to refer to the first set of dimensions. In line 181, remove the word experiment, as this study was not an experiment.
Table 2: clarify the meaning of the bracketed numbers in the ‘Mean’ column. Later on, in the discussion, these numbers represent the points scored for the domains, but I could not see how these points were derived.
When using Likert scales, it is often better to use medians and interquartile rankings. Doing so, would give a better sense of where the scores sit, i.e. whether in the low, medium or high bands of each domain. There may be very good reasons why you chose to use Means, and if so, please explain.
In section 3.2 you have highlighted the statistically significant differences, and it would be useful to include the direction of this difference within the text. For example there was a statistically significant difference between men and women, with men being more stressed.
In section 3.2.3 you refer to results that positively predicted stress and other behaviours and negatively predicted stress. It may be useful to clarify or re-word ‘negatively predicted’.
The discussion raises some interesting points, however, it is unlikely that with a sample size of n = 168, that some of the assumptions can be generalized, especially as there was no mention of a power calculation having been carried out.
Later in the discussion (lines 361 on) you refer to a 1:4 caregiver to patient ratio, and a few is also mentioned. This needs to be clarified for the international readership. What this sentence suggests, is that a care-giver (a family member of one patient), looks after four patients (their own family member plus three others), whilst the patients are in hospital, and they pay a fee to do this. Is this the case? If so, this is not a model of care that I (or others) will be familiar with, and thus, it would need to be clarified and explained. Later on, you refer to professional care-givers. Who are these, and where are the healthcare staff (nurses doctors etc.)?
When you refer to ‘sometimes’, ‘often’ etc. (lines 431 onwards), this does not really give an indication of the meaning. As mentioned above, there needs to be an example of the context of these domains. What were your participants asked? This section also refers to patients’ coping, rather than care-givers, as in your study, and I don’t think that this is the same thing.
The discussion would benefit from being written more succinctly. For example, the section from line 470, is repetitive in places.
The conclusion brings in a number of previously undiscussed concepts, including speculation and recommendations around nursing staff, and their roles (e.g. line 629). There does not seem to be evidence for many of these claims, and so I suggest that the conclusions are kept very short, and reflect the size and type of this study.
Ans:
(1) Thank you for your kind comments on our manuscript. We have explained why additional care-giving is required when the older person is hospitalized and what a cooperation bed mean in the manuscript, as outlined on page 2, lines 86-90.
(2)Thank you for your comment, and our reply is as follows: The family function referred to in the article pertains to caregiver.
(3)Thank you for your comment, and our reply is as follows: A quantitative, cross-sectional survey method in this article. We conducting face-to-face distribution of questionnaires did not involve extensive interviews.
(4) The participants in this study were family caregivers of residents hospitalized in long-term care facilities from August 2021 to April 2022. This has been added to lines 109-111. Supplementary explanations for each aspect of the scale and the meaning of scores are also stated more clearly. For details, see Line 122-132.
(5) Thank you for your kind comments on our manuscript. We have additional different mean by problem-focused coping behaviors and emotion-focused coping behaviors in the manuscript, as outlined on page 15, lines 459-463.
(6)Thank you for your kind comments on our manuscript. We have corrected to correct data it was fitted with sentence beginning in line 172 and remove the word experiment correct to research.
(7)Thank you for your kind comments on our manuscript. We have added clarify the meaning of the bracketed numbers in the “Mean” column, as outlined on page 5, Table 2.
(8)Thank you for your kind comments on our manuscript. Acutally, we used Likert scaels to better sense of where the scores sit. Likert scaels have five grade to understand more about the actual situation of participants.
(9) Thank you for your correction. This study is a purposive sampling. It can indeed only reflect the situation of a hospital in Changhua, and no inferences can be made. It has been stated in the research limitations.
(10) Thanks for the correction. The researcher has made a more concise statement of the conclusion.
(11) Thank you for your kind comments on our manuscript. We have
included additional more in-depth explaintion wha is the “cooperation bed” mean in the manuscript, as outlined on page 2, lines 86-90. To avoid misunderstanding by the readers, "nursing beds" has be changed to "cooperative beds."
The limitations were reasonable, as were the recommendations.
Please proof-read the work for English language Editing checks:
Lines 12 and 13: check for clarity
Ans:
(1)Thank you for the correction. We want to convey that the care of the elderly is originally provided by nursing homes, but if the patient is hospitalized, the caregiving burden will shift back to the family.
Line 18: Please do not use the word ‘object’ for people. I suggest you use participant or another more person-centered term
Ans:
(1)Thank you for the correction. We have already translated "research object " to "research participant ".
Line 27: correct the spelling of significant and
Ans:
(1)Thank you for the correction. We have already corrected the spelling.
Line 38: I am not sure that reach is the correct word.
Ans:
(1)Thank you for the correction. We have already translated "representing ratio"
Lines 48+: Please use the future tense, where applicable
Ans:
(1)Thank you for the correction. We have already made the corrections in the future tense
Line 67: Please use more person-centered words. Demented individuals is not person-centered. Perhaps older people living with disabilities and/or dementia.
Ans:
(1)Thank you for the correction. We have already modified “for older people living with disabilities or dementia”
Line 75: Do you mean papers? Perhaps studies might work as well?
Ans:
(1)Thank you for the correction. We have already translated "papers" to "studies"
Line 152: Performed
Ans:
(1)Thank you for the correction. We have already corrected the spelling.
Lines 155, 212+: Please use participants rather than subjects.
Ans:
(1)Thank you for the correction. We have already translated "subjets" to "participants"
Line 157: Include the unit of currency, and it might be useful to add a USD equivalent, or a reference to the mean income, for context?
Ans:
(1)Thank you for the correction. I have added USD equivalent.

Round 2
Reviewer 2 Report
Comments and Suggestions for Authors
It is not clear how the 4 subscales were identified with the stress instrument, content validity is mentioned however there is no mention of factor analysis. The Cronbach is low for the life subscale with the stress instrument-how many ties has the stress instument been used?
Comments on the Quality of English LanguageEditing is required, there are run on sentences and misuse of tense throughout the paper
Author Response
Dear Reviewer 2,
Thank you to the editors and reviewers for taking the time to review our manuscript.
We also appreciate for your valuable comments and advice. These comments are all valuable and very helpful for revising and improving our paper. We have revised the manuscript accordingly, and the detailed point-by-point responses are listed below. Additionally, we have provided explanations and supplements addressing each of the comments.
Reviewer 2:
It is not clear how the 4 subscales were identified with the stress instrument, content validity is mentioned however there is no mention of factor analysis. The Cronbach is low for the life subscale with the stress instrument-how many ties has the stress instument been used?
Ans:
(1)The four dimensions of the research tool were derived from a literature review. In this study, only reliability and validity analyses were conducted, and factor analysis was not performed. Future research will consider incorporating this suggestion.
Reviewer 3 Report
Comments and Suggestions for Authors
Dear Authors,
Thank you for addressing my comments.
Looking again at Table 2, I do not understand the Mean/Sum column. Is the mean with or without brackets? If without, how can the mean be lower than the range (e.g. economy range = 4 - 20 yet the mean is 2.18). If the mean is in brackets (which makes more sense), then I do not uinderstand the relevance of the other numbers.
Please clarify the content and labelling, and address the very few minor editing errors (final proof-reading needed) and I wish you success in your ongoing work.
Comments on the Quality of English LanguageA few minor editing errors, so further proof-reading is needed. For example, line 481: females' (plural possessive); line 484: females (plural); line 661: Explore
Author Response
Dear Reviewer 3,
Thank you to the editors and reviewers for taking the time to review our manuscript.
We also appreciate for your valuable comments and advice. These comments are all valuable and very helpful for revising and improving our paper. We have revised the manuscript accordingly, and the detailed point-by-point responses are listed below. Additionally, we have provided explanations and supplements addressing each of the comments.
Reviewer3:
Looking again at Table 2, I do not understand the Mean/Sum column. Is the mean with or without brackets? If without, how can the mean be lower than the range (e.g. economy range = 4 - 20 yet the mean is 2.18). If the mean is in brackets (which makes more sense), then I do not uinderstand the relevance of the other numbers.
Comments on the Quality of English Language
A few minor editing errors, so further proof-reading is needed. For example, line 481: females' (plural possessive); line 484: females (plural); line 661: Explore
Ans:
(1)Thank you for your kind comments on our manuscript. We have reorganized the Mean/Sum column. The Sum without brackets, the mean with brackets. The range can be used to explain the sum.
(2)Thank you for the correction. We have already corrected the spelling in line 481,484,661.